# Kynureninase Promotes Immunosuppression and Predicts Survival in Glioma Patients: In Silico Data Analyses of the Chinese Glioma Genome Atlas (CGGA) and of the Cancer Genome Atlas (TCGA)

**DOI:** 10.3390/ph16030369

**Published:** 2023-02-28

**Authors:** Gonzalo Pérez de la Cruz, Verónica Pérez de la Cruz, Javier Navarro Cossio, Gustavo Ignacio Vázquez Cervantes, Aleli Salazar, Mario Orozco Morales, Benjamin Pineda

**Affiliations:** 1Department of Mathematics, Faculty of Sciences, Universidad Nacional Autónoma de México, UNAM, Mexico City 04510, Mexico; 2Neurobiochemistry and Behavior Laboratory, National Institute of Neurology and Neurosurgery “Manuel Velasco Suárez”, Mexico City 14269, Mexico; 3Neuroimmunology Unit, National Institute of Neurology and Neurosurgery “Manuel Velasco Suárez”, Mexico City 14269, Mexico

**Keywords:** tryptophan catabolism, kynureninase, glioblastoma, immunosuppression

## Abstract

Kynureninase (KYNU) is a kynurenine pathway (KP) enzyme that produces metabolites with immunomodulatory properties. In recent years, overactivation of KP has been associated with poor prognosis of several types of cancer, in particular by promoting the invasion, metastasis, and chemoresistance of cancer cells. However, the role of KYNU in gliomas remains to be explored. In this study, we used the available data from TCGA, CGGA and GTEx projects to analyze KYNU expression in gliomas and healthy tissue, as well as the potential contribution of KYNU in the tumor immune infiltrate. In addition, immune-related genes were screened with KYNU expression. KYNU expression correlated with the increased malignancy of astrocytic tumors. Survival analysis in primary astrocytomas showed that KYNU expression correlated with poor prognosis. Additionally, KYNU expression correlated positively with several genes related to an immunosuppressive microenvironment and with the characteristic immune tumor infiltrate. These findings indicate that KYNU could be a potential therapeutic target for modulating the tumor microenvironment and enhancing an effective antitumor immune response.

## 1. Introduction

Glial cells constitute approximately 50% of the brain cell population in proportion to neural cells [1]; transformed glial cells give rise to gliomas, which are the most common primary brain tumors [2]. Among gliomas, glioblastoma (GBM) is the most aggressive and frequent tumor arising from transformed astrocytes, with an incidence rate of 3.2 per 100,000 [3,4]. Despite the current standard of treatment established by Stupp since 2005 for GBM patients, consisting of radiotherapy and temozolomide, mean survival remains close to 14.6 months. This prognosis has not changed in the last decade; thus, GBM is considered an incurable illness [5]. GBM’s poor prognosis has been attributed mainly to its high biological complexity, which is insufficiently understood [6]. The multiple genetic alterations in GBM confer several biological advantages, such as treatment resistance, high proliferation and metabolic adaptations that allow modulation of the tumor microenvironment (TME) [7]. In this context, mutations in the active site of isocitrate dehydrogenase 1 (IDH1), a catalytic enzyme that produces alpha-ketoglutarate (α-KG) by oxidative decarboxylation of isocitrate [8], have been related to a better prognosis in glioma patients. Specifically, IDH1 gene mutation in the codon 132 is a frequent mutation present in low-grade gliomas (around 80%) and secondary glioblastomas. The prevalence of IDH1 mutation predicts the origin of low-grade gliomas, as most of them have this mutation; on the contrary, primary glioblastomas have a low frequency of the mutated IDH1 gene [9].

The GBM tumor microenvironment is highly heterogeneous and consists of different cell types involved in promoting immune evasion [10]. Moreover, a close relationship between metabolic reprogramming and immunosuppressive TME has been observed [11], with immune evasion being the main hallmark described in GBM [6]. Thus, alterations in metabolic pathways can promote an immunosuppressive TME through the release of different oncometabolites [12]. In this context, several reports showed that gliomas exhibit a high consumption of tryptophan (Trp), thus possibly promoting a state of immunosuppression by reducing Trp bioavailability, which could induce anergy of effector T cells. In addition, this high Trp consumption by GBM can promote the release of several oncometabolites produced during its degradation through the kynurenine pathway (KP) (Figure 1).

The Trp oncometabolites promote Treg activation and the suppression of local CD8+ T and natural killer (NK) cells, as well as a reduced humoral immune response favoring tumor invasion and progression [13,14]. KP represents about 95% of Trp degradation to bioactive metabolites known as kynurenines, further producing nicotinamide adenine dinucleotide (NAD+) [15]; the remaining Trp is used for serotonin production and protein synthesis. In addition, the Trp-degrading enzyme indoleamine-2,3-dioxygenase 1 (IDO1), which is a widely expressed and interferon-inducible enzyme, produces kynurenine (KYN) and is highly expressed in multiple types of human cancers, including GBM [16,17,18,19]. Notably, KYN is an endogenous ligand for the aryl hydrocarbon receptor (AhR), a xenobiotic-responsive transcription factor that has been implicated in tumor growth and motility by its dimerization with the aryl hydrocarbon nuclear translocator (ARNT) that regulates a wide variety of xenobiotics genes [20]. In addition, IDO1 expression has been associated with an immunosuppressive TME, infiltration of tumor-associated macrophages (TAMs) and regulatory T lymphocytes (Tregs), which results in a poor survival rate in glioma patients [21,22,23]. In preclinical studies, IDO1 inhibitors have shown a low tumor burden, and when combined with immune checkpoint therapy, such as anti-PD-L1 and CTL4, tend to provide a synergistic benefit by reducing brain tumor mass and the amount of tumor-infiltrating Tregs while increasing the survival rate of malignant glioma-bearing mice [24]. However, the successful use of IDO inhibitors in clinical studies for GBM has not been reported. The ECHO-301 confirmatory phase 3 study in patients with advanced melanoma was designed to determine the effectivity of epacadostat (an IDO/TDO inhibitor) in combination with pembrolizumab (anti PD-1 antibody); this combination showed a better overall response rate than that observed with anti-PD-1 alone. However, epacadostat failed to improve clinical responses [25]. Although the components of the KP have been associated with poor prognosis in various types of cancer, including GBM, most research has focused on the study of IDO, mainly because little is known of the expression of KP enzymes in various cell types, as well as the physiological function of the KP metabolites produced along this pathway. So far, KP components have been poorly studied in GBM, despite the fact that metabolites produced from Trp degradation have demonstrated immunosuppressive properties that could be involved in tumor progression [26,27,28]. Relatedly, a recent study showed that different GBM cell lines can produce kynurenic acid (KYNA), the short arm of KP, and metabolites from the long arm of this pathway, such as 3-hydroxykynurenine (3-HK) and 3-hydroxyanthranilic acid (3-HANA) [29,30]. Interestingly, the kynurenine monooxygenase (KMO) enzyme, which is not normally expressed in healthy astrocytes, was found to be expressed and functional in human astrocytomas [29], thus suggesting that other enzymes besides IDO could contribute to the production of KP-derived oncometabolites that can promote an immunosuppressive microenvironment in GBM and, consequently, decrease the overall survival of these patients.

In this regard, kynureninase or L-kynurenine hydrolase (KYNU) is a KP enzyme located in 2q22.2 gene that has been poorly studied in glioma. KYNU has been largely related with several pathologies, such as metabolic neurological, cardiac and renal diseases [31,32,33]. This enzyme can use KYN or 3-HK as a substrate to produce anthranilic acid (AA) and 3-HANA, respectively [34]. Specifically, 3-HANA has been described as a modulator of the immune response, since 3-HANA can induce apoptosis, potentiated by ferrous or manganese ions, in monocytes and macrophages under inflammatory conditions [35], as well as triggering the activated T cell death by depletion of intracellular GSH [36]. Furthermore, 3-HANA inhibits the maturation of dendritic cells and suppresses T cell stimulation [37]. A recent report showed that KYNU expression correlated with poor overall survival in lung adenocarcinoma, by promoting tumor immunosuppression through the induction of tumor-infiltrated Tregs accompanied by a concordant increase of PD1 and PD-L1 protein levels [38]. Recently, KYNU has gained attention in several pathologies due to its immunomodulatory ability [39]. Specifically, KYNU is highly upregulated in the HER2-enriched and triple negative breast cancer subtypes [40], suggesting that inhibitors of KYNU could represent an alternative therapeutic target. Additionally, KYNU is highly overexpressed, both in cell lines and tumor tissue of cutaneous squamous cell carcinoma (cSCC). Lastly, the proliferation, migration and invasiveness of cSCC cell line are suppressed when KYNU is silenced, suggesting that KYNU could take a key role in the malignant progression of cSCC [41].

Considering these previous studies which indicated that KYNU could be a key KP enzyme in cancer, we decided to analyze the expression and the biological function of KYNU in gliomas, using the data available in the Chinese Glioma Genome Atlas (CGGA) and The Cancer Genome Atlas (TCGA).

## 2. Results

### 2.1. KYNU Expression on Healthy Tissue and Tumor Tissue

A gene expression profiling interactive analysis (GEPIA) website was used to explore the expression pattern of KYNU on healthy conditions (based on the genotype tissue expression, GTEx, database) and tumoral tissue (based on TCGA database) for different cancers; see Figure 2. KYNU is overexpressed in a variety of tumors, including cervical squamous cell carcinoma and endocervical adenocarcinoma, lymphoid neoplasm diffuse large B-cell lymphoma, esophageal carcinoma, head and neck squamous cell carcinoma, ovarian serous cystadenocarcinoma, pancreatic adenocarcinoma, skin cutaneous melanoma metastasis and stomach adenocarcinoma when compared to control peritumoral tissues (Figure 2A,B). Specifically, the analysis showed that the normal brain tissue expresses a low amount of KYNU mRNA compared to tumor brain tissue (Figure 2A). In addition, tumoral tissue from both low-grade glioma (LGG) and GBM-overexpressed KYNU when compared to healthy brain tissue expression (around 2-fold and 4-fold, respectively) (Figure 2B).

### 2.2. KYNU Expression in Glioma Using the Chinese Glioma Genome Atlas (Cgga)

Once we showed KYNU overexpression in some cancers using the TCGA-GTEx data, especially in GBM and LGG, the next step was to evaluate KYNU expression in glioma using both TCGA and CGGA data (Figure 3A,B, respectively) to determine if these alterations in the expression of KYNU were also present in another population group (different race). When considering TCGA data, KYNU expression was significantly increased in LGG and GBM compared to control tissue (Figure 3A); however, the expression of this enzyme significantly differed between GBM and LGG, being higher in the most aggressive tumor. In the case of CGGA data, KYNU expression on GBM was also significantly higher when compared to LGG (Figure 3B); in this case, it is worth mentioning that it was not possible to compare with control tissue, since the CGGA database does not have information on normal tissue. However, both datasets showed the same trend concerning KYNU expression in glioma.

Then, the astrocytoma samples from the CGGA were divided according to the World Health Organization’s (WHO) classification of tumors (Figure 4). It is evident that the greater the tumor grade, the higher the expression of KYNU (Figure 4A). When the IDH mutation status (mutant vs. wildtype) was considered, the KYNU expression was higher in the IDH-wildtype group (Figure 4B); and when the tumor grade was considered together with the IDH wildtype status, KYNU expression was significantly higher in the wildtype group in all tumor grades (Figure 4C–E).

### 2.3. Prognostic Value of KYNU Expression in Primary Astrocytomas

Overall survival (OS) analysis from TCGA and CGGA datasets in low-grade gliomas showed that KYNU expression correlated with poor prognosis (*p* < 0.0001) (Figure 5A,B); that is, the higher the expression of KYNU, the lower the probability of survival. For GBM, KYNU has prognostic value, showing that the higher the expression of KYNU, the lower the OS (*p* < 0.0324), when only considering the TCGA data (Figure 5A).

### 2.4. KYNU Expression Correlates with Immune Mediators Related to Immunosuppression

Considering that KYNU is overexpressed in astrocytoma and has a prognostic value in OS, the next step was to analyze the association of KYNU with immunosuppression. Thus, various well-described immune mediators in astrocytoma were screened together with KYNU. Co-expression analysis of IL2R, ITGAM, MMP9, PDCD1, PDCD1LG2, IDH1, HLA-E, CD47, IL10, FAS, TGFβ, CCL2, CCL4 and CTL4 revealed that all these mediators were positively correlated with KYNU (Figure 6) in both datasets.

### 2.5. KYNU Expression Correlates with the Immune Infiltrate

The correlation analysis between KYNU and immune mediators indicated that the genes in the KYNU high-expression group were enriched in immune-related pathways. Thus, the immune infiltration of GBM and LGG was evaluated. The cell type-level expression analysis showed that effector populations of T cytotoxic CD8+ cells, NK-activated cells and M1 macrophages are reduced in LGG and GBM, while regulatory M2 macrophages are increased in comparison with normal tissue (Figure 7A). When KYNU expression was considered with the cell infiltrated with GBM, we found that KYNU is overexpressed in cells with immunosuppressive activity, such as myeloid dendritic cell (MDC), Tregs and M2 macrophages (Figure 7B). In the case of LGG, KYNU is overexpressed in macrophages M2.

When myeloid dendritic cells were analyzed, we found that KYNU is also overexpressed in this cell sub-population, both in the GBM and LGG group, when compared to the control group (normal) (Figure 8). The changes in KYNU copy number appeared to significantly influence the immune infiltration level in GBM.

## 3. Discussion

GBM is a highly aggressive and lethal brain neoplasia; it is considered a cold tumor with poor response to immunotherapies, due to its intrinsic characteristics that promote an immunosuppressive TME [42]. In addition to immune evasion, metabolic alterations are one of the most important hallmarks of GBM [43]. KP has been recognized as an important metabolic contributing factor for inducing immunosuppression in GBM, due to the oncometabolites that are produced, promoting Treg differentiation, increasing the production of anti-inflammatory cytokines and, consequently, inducing low cytotoxic activity of T lymphocytes [7]. A previous bioinformatic analysis that focused on the relationship between gliomas hallmarks and KP demonstrated a close relationship between the KP enzymes and several genes involved in angiogenesis, Signal Transducer and Activator of Transcription (STAT) signaling, Rho GTPases and, mainly, with genes related to immune response modulation [44]. Therapeutic potentiation of the immune system through KP modulation is an attractive strategy to improve current GBM treatment. For this reason, multiple phase 1/2 trials were focused on modulating IDO (the first and rate-limiting step of the catabolic tryptophan-kynurenine pathway frequently overexpressed in GBM) through small molecule inhibitors to improve the patient response to immune checkpoint inhibitors, such as anti-PD-1 and anti-PD-L1 therapy. However, these trials have not provided survival benefits to date [21,25,45]. Considering that KP metabolites downstream IDO are related to the immune modulation and several glioma hallmarks, this study was focused on understanding the contribution of KYNU in GBM and how it influences the anti-tumor immune response and consequently the survival of glioma patients through a bioinformatic approach.

To note, KYNU is highly expressed in healthy tissues, such as the lungs, urinary bladder, appendix and liver, mainly because KP is very active in these organs where tryptophan dioxygenase (TDO) is the principal enzyme involved in Trp degradation [46]. In contrast, KYNU expression is low in brain tissue, where IDO is the main enzyme responsible for catalyzing the conversion of Trp to KYN in microglial cells, particularly in an inflammatory context [34]. Some strategies target both TDO and IDO1 enzymes since, when IDO1 is inhibited, TDO together with IDO2 could exert a compensatory effect on KYN production, favoring immunosuppression [47]. On the other hand, it was shown herein that, when the tumor appears, KYNU brain expression increased more than 4-fold, which is in accordance with a previous report in which KP enzymes are overexpressed in brain tumor pathogenesis [48]. Since KYNU is normally expressed in all brain cell types [34], its higher expression in brain tumors could be due to the mutation status in astrocytes or because of the increase in the number of tumor-infiltrating cells, mainly phagocytic cells, which are known to account for up to 30% of the tumor mass in GBM [49,50]. The increased expression of KYNU suggests that this enzyme plays a potential role in the pathogenesis of GBM, as well as in other tumors where this enzyme is overexpressed when compared to its healthy counterpart (Figure 2A). Additionally, KYNU expression was negatively correlated with overall survival in both LGG and GBM, indicating that KYNU might be an effective marker to predict the prognosis of glioma patients, regardless of race. In this context, it was suggested that KYNU could be used as a potential prognostic factor in lung adenocarcinoma with activated NRF2, since it was correlated with Treg maturation, increased levels of PD-1 and PD-L1, immunosuppression and, consequently, poor overall survival [38]. Additional studies showed that high levels of 3-HANA (a product of 3-HK degradation catalyzed by KYNU) were associated with poor progression-free survival in non-small cell lung cancer patients [51].

Furthermore, when KYNU expression was analyzed in gliomas, a higher KYNU expression was found, dependent on the grade of malignancy and regardless of whether the tumor was primary or recurrent. A similar effect was found when KYNU expression was analyzed in astrocytomas with different WHO tumor grades (Figure 4). In addition, KYNU expression was analyzed depending on the IDH mutation, which is known as a prognostic factor for brain tumor patients [52,53]; under this condition, the wildtype IDH status showed the higher KYNU expression, correlating with poor survival outcome in GBM patients. In the case of LLG, the IDH1 mutation status has been associated with a more favorable outcome; it is worth mentioning that this mutation occurs at an early stage during malignant transformation, promoting gliomagenesis [54,55]. The IDH wildtype enzyme uses nicotinamide adenine dinucleotide phosphate (NADP+) as a cofactor, to convert isocitrate to α-KG and NADPH. However, IDH1-mutated enzymes convert α-KG to 2-hydroxyglutarate, an oncometabolite that induces stem cells’ differentiation into gliomas by inhibition of α-KG-dependent enzymes [8]. Thus, KYNU might be a key factor in promoting gliomagenesis; however, there are no reports describing the potential association between IDH status and alterations in KYNU expression. The fact that KYNU expression was negatively related to poor survival outcomes in GBM suggests that KYNU inhibition might be an effective strategy to improve the prognosis of GBM patients. As for LGG, it could prevent or extend the time of tumor progression in different ways, such as the potential regulation of IDH, reducing the production of reactive oxygen species or through the regulation of epigenetic modifiers; however, these possibilities should be further explored in future research. Nonetheless, it is well known that KP is highly interconnected with several molecular pathways related to carcinogenesis in GBM, including phosphoinositide-3 kinase (PI3K), extracellular-signal-regulated kinase (ERK), Wnt/β-catenin and p53, among others [56]. According with our findings, previous reports showed that high KYNU expression correlates with poor overall survival in other cancers, such as lung adenocarcinoma (LUAD) [38,57] and breast cancer [40,58]. Additionally, KYNU was associated with the promotion of cellular growth of cutaneous squamous cell carcinoma cells [41].

During the development of brain tumors, several tumor-associated cells infiltrate the tumor mass, promoting an immunosuppressive TME [50]. In this line, KP metabolites could play an important role, due to their immunomodulatory functions. Recent studies have shown that KYN and KYNA are endogenous AhR agonists [59,60], whose activation induces the differentiation of Treg cells and an immune tolerance response, preventing the elimination of tumor cells [61,62]. Furthermore, it was observed that the nuclear coactivator 7, which is highly expressed in dendritic cells, is a molecular target of 3-HANA. This oncometabolite favors the interaction of nuclear coactivator 7 with AhR and increases AhR-dependent gene transcription, thus potentiating their immunomodulatory effects [63]. AhR activation by kynurenines could induce the transcription factor FoxP3, promoting the inhibition of cytotoxic effector T cells, such as CD8+ T cells and NK cells, thus allowing tumor cells to escape destruction [64,65]. Additionally, 3-HANA mediated the T cells’ apoptosis by the activation of PDK1 kinase and caspase 8 [66,67]. In light of this, it is possible that the effect of KYNU on the poor prognosis of several cancers could be partially dependent on its catabolic product 3-HANA, since this oncometabolite appears to be relevant in the modulation of the antitumor immune response. Moreover, KYNU expression in human astrocytomas confers several advantages that are associated with 3-HANA toxicity, whereas this oncometabolite promotes a prooxidant environment, altering the functionality of T effector cells, thus enhancing the immunosuppressive TME [68]. Therefore, as this analysis suggests, KYNU could promote the infiltration of immune cells, mainly in regulatory cells where KYNU expression correlated significantly and positively with chemoattractant molecules, such as monocyte chemoattractant protein-1 (MCP-1/CCL2), macrophage inflammatory protein-1 beta (MIP-1β/CCL4) and with the expression of IL2R and ITGAM. Additionally, cell type-level expression analysis showed that KYNU expression is closely related to myeloid dendritic cells, macrophage M2 and Treg cell levels, and could predict the prognosis of patients with astrocytic tumors. Additionally, KYNU expression positively correlated with the expression of immunoregulatory cytokines (IL-10 and TGF-β) and immune checkpoints inhibitors, such as PD-1 and PDL-1 and CTLA-4, as well as with matrix metalloproteinase 9 (MMP-9), an enzyme involved in the breakdown of the extracellular matrix that promotes tumoral invasion in GBM and is correlated with poor prognosis [69]. According to our results, other studies in LUAD revealed that KYNU expression is associated with an immunosuppressive TME, with an increase in the number of Tregs in the tumor mass and an increase in the amount of PD-1 and PD-L1 protein [38]. Moreover, a recent study proposed that KYNU could be a potential transcriptional target of CD44-downstream signaling promoting tumor cell invasion, cellular migration and survival, and metastasis in breast cancer [58]. The findings derived from this study showed that KYNU is a target enzyme positively associated with immunosuppression and may be a prognostic marker in glioma. Future studies by our group will aim to elucidate the precise mechanism by which this enzyme modulates the tumoral microenvironment favoring tumor growth.

## 4. Materials and Methods

### 4.1. Data Acquisition and Sample Selection

Two datasets were considered. The first corresponds to gene expression RNAseq (RSEM tpm) and clinical data from a combined cohort of TCGA, TARGET and GTEx samples, downloaded from the Xena platform [70]. This dataset contains information from 19,131 samples, but only those from TCGA and GTEx projects were selected. Detailed analysis of three groups was based on the samples associated with categories: TCGA brain low-grade glioma (LGG, n = 523); TCGA glioblastoma multiforme (GBM, n = 166); and the union of GTEx brain cortex and GTEx brain frontal cortex (Ba9) (Normal: n = 105 and n = 102, respectively).

The second dataset corresponds to gene expression RNAseq (RSEM fpkm) and clinical data downloaded from the Chinese Glioma Genome Atlas (CGGA, mRNAseq_693 dataset) [71]. From a total of 693 samples, only those corresponding to grades WHO II (n = 188), WHO III (n = 255) and WHO IV (n = 249) were selected.

### 4.2. Selection of Gene Expressions

Kynureninase (KYNU) expression and the following group of genes associated with immune response were selected: Interleukin 21 Receptor (IL21R), Integrin Subunit Alpha M (ITGAM), Matrix metalloproteinase 9 (MMP9), Programmed Cell Death 1 (PDCD1), Programmed cell death 1 ligand 2 (PDCD1LG2), Isocitrate dehydrogenase 1 (IDH1), human leukocyte antigen E (HLA-E), integrin-associated protein (CD47), interleukin-10 (IL10), CD95 (tumor necrosis factor receptor superfamily member 6, FAS), Transforming Growth Factor Beta 1 (TGFB1), monocyte chemotactic protein (CCL2), chemokine (C–C motif) ligand 4 (CCL4) and Cytotoxic T lymphocyte antigen-4 (CTLA4). Indoleamine dioxygenase (IDO1) was also considered.

### 4.3. Statistical Analysis

The Kruskal–Wallis test and Dunn’s test for multiple pairwise comparisons were used to evaluate the difference in KYNU expression among groups of samples (e.g., GBM, LGG and Normal).

For each of the two groups, GBM and LGG, Spearman’s correlations, with the Holm adjustment for multiple tests, were performed to simultaneously analyze the pairwise correlations of the KYNU expression and the expression of each of the genes associated with immune response.

A Cox proportional hazards regression model was used to assess whether KYNU expression has an effect on survival probability. The model included a stratification defined by the two groups of samples (GBM and LGG), one continuous covariate (the expression levels of KYNU), and the interaction between this continuous covariate and the categorical variable with two levels (GBM and LGG). The likelihood ratio test, the Wald test and the score (log-rank) test were performed to assess whether KYNU expression has an effect in at least one of the two groups, GBM and LGG. Then, tests with adjusted *p*-values for simultaneous inference were used to distinguish the individual groups where the effect is statistically significant.

A *p*-value less than 0.05 was considered statistically significant, and R v4.0.2 (Vienna, Austria) [72] was utilized to generate the results and Figure 3, Figure 4, Figure 5 and Figure 6 using the packages survival (v3.1-12) [73], multcomp (v1.4-15) [74] and psych (v2.0.9) [75].

Figure 2, on the median expression of KYNU, was generated using the GEPIA2 tool [76] considering the General Information option. Figure 7 and Figure 8 on the infiltration level of immune cells (myeloid dendritic cells, T cells CD8+, Tregs, activated NK cells, macrophages M1 and macrophages M2) were obtained using the GEPIA2021 tool [77], considering the options Proportion Analysis and Sub-expression Analysis for KYNU, and in both cases selecting CIBERSORT for the deconvolution, except for myeloid dendritic cells, since it was only available selecting quanTIseq. Comparisons among GBM, LGG and Normal samples at the cell type-level were performed using one-way ANOVA. GEPIA2 and GEPIA2021 are based on the TCGA and GTEx samples from the Xena platform.

## 5. Conclusions

This study highlights that the overexpression of KYNU is closely related to the immune infiltration level and represents a prognostic marker for GBM and LGG patients. Further investigations exploring how KYNU impacts antitumor immune response, and the potential targetability of KYNU as an ‘immuno-metabolic’ adjuvant in astrocytoma that could improve the current immunotherapeutic approaches in the arsenal of GBM and LGG treatments, are needed.

## Figures and Tables

**Figure 1 pharmaceuticals-16-00369-f001:**
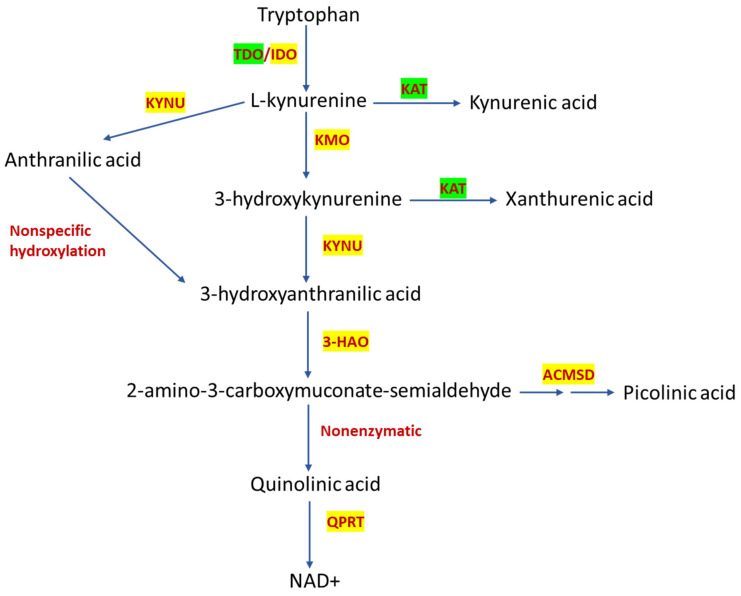
Kynurenine pathway. Tryptophan catabolism through KP leads to nicotinamide adenine dinucleotide (NAD+) formation. The first step is performed by tryptophan dioxygenase (TDO) in the liver or by indoleamine dioxygenase (IDO) in the brain to produce L-kynurenine (L-KYN). L-KYN is a substrate for three enzymes: kynurenine aminotransferase (KAT) to produce kynurenic acid (KYNA); kynureninase (KYNU) to produce anthranilic acid (ANA); and kynurenine monooxygenase (KMO) to produce 3-hydroxykynurenine (3-HK). Then, KYNU takes 3-HK as a substrate to produce 3-hydroxyanthranilic acid (3-HANA), which can also be metabolized by a nonspecific hydroxylation of ANA. 3-hydroxyanthranilate oxidase (3-HAO) opens the ring of 3-HANA, producing the unstable product 2-amino-3-carboxymuconate-semialdehyde, further metabolized to produce picolinic acid (PIC) through α-amino-β-carboxymuconate-semialdehyde-decarboxylase (ACMSC) or quinolinic acid (QUIN) by a nonenzymatic reaction. Finally, quinolinate phosphoribosyltransferase (QPRT) leads to NAD+ production. The overexpressed KP enzymes in glioma are in yellow and down-expressed enzymes are in green.

**Figure 2 pharmaceuticals-16-00369-f002:**
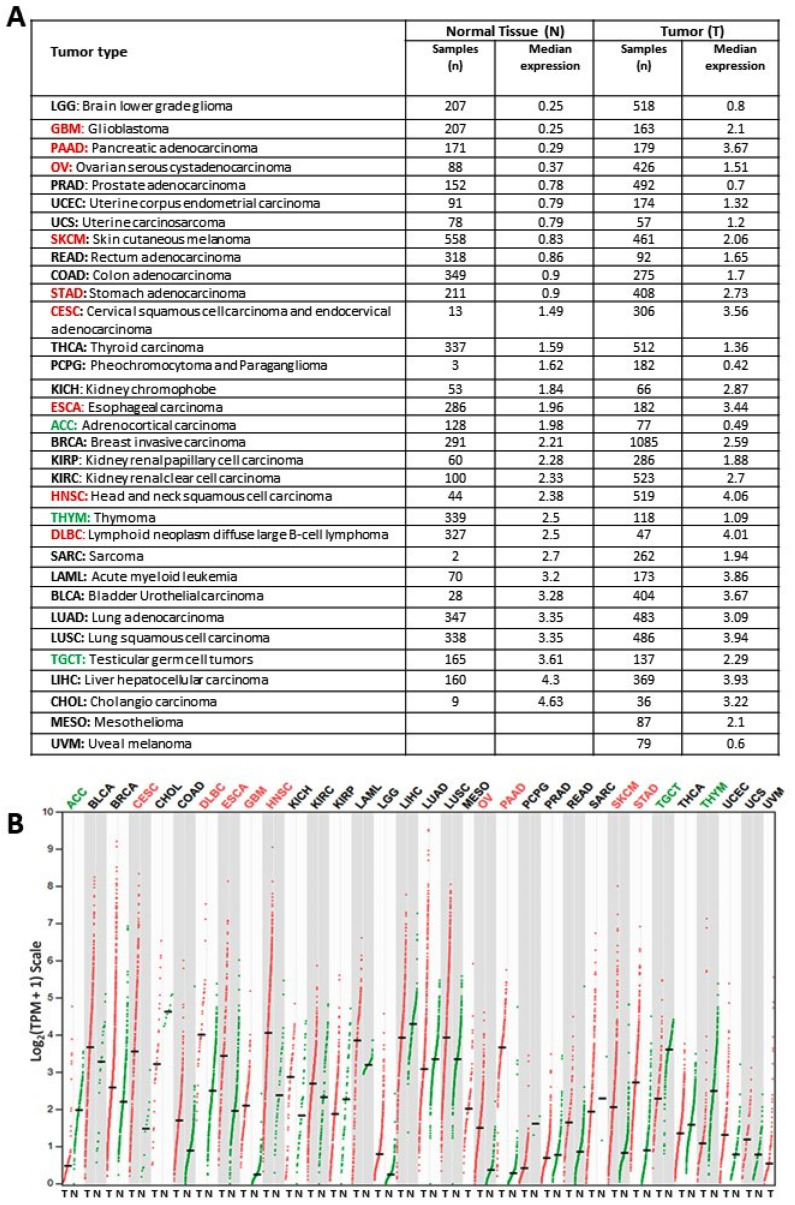
KYNU expression using TCGA tumor samples (T, red) and corresponding TCGA-GTEx normal tissues (N, green) for different tumors: (**A**) the KYNU median expression and (**B**) dot plot, where each dot represents the KYNU expression of a sample, and the black horizontal line represents the median expression.

**Figure 3 pharmaceuticals-16-00369-f003:**
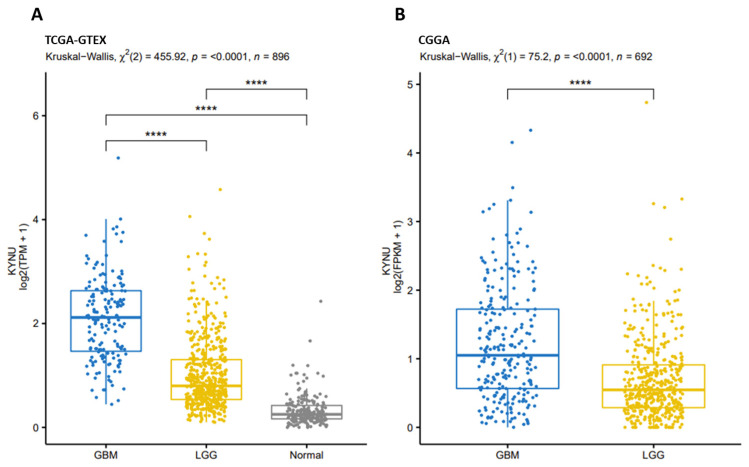
KYNU expression in each group of samples: (**A**) GBM (TCGA-GBM), LGG (TCGA-LGG) and Normal (GTEx-Brain cortex) and (**B**) GBM (CGGA-WHO IV) and LGG (CGGA-WHO II and WHO III). **** *p* < 0.0001; Dunn’s test for multiple pairwise comparisons.

**Figure 4 pharmaceuticals-16-00369-f004:**
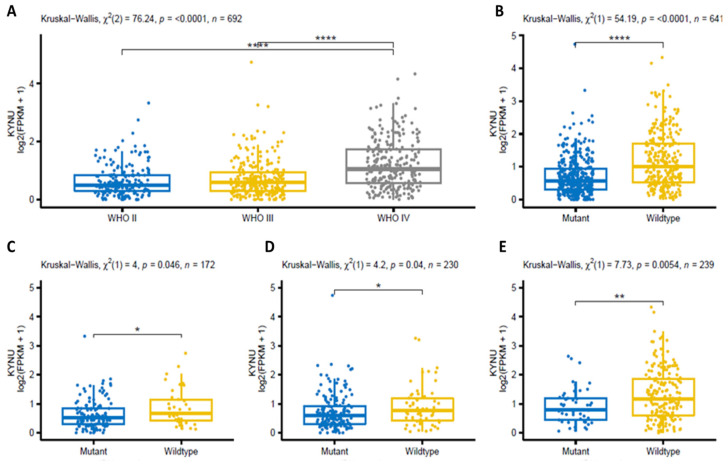
KYNU expression in the CGGA dataset for the groups of samples associated with (**A**) WHO classification of tumors (grades II to IV), (**B**) IDH mutation status (mutant and wildtype) and (**C**–**E**) IDH mutation status in each WHO tumor grade. * *p* < 0.05, ** *p* < 0.01 and **** *p* < 0.0001; Dunn’s test for multiple pairwise comparisons.

**Figure 5 pharmaceuticals-16-00369-f005:**
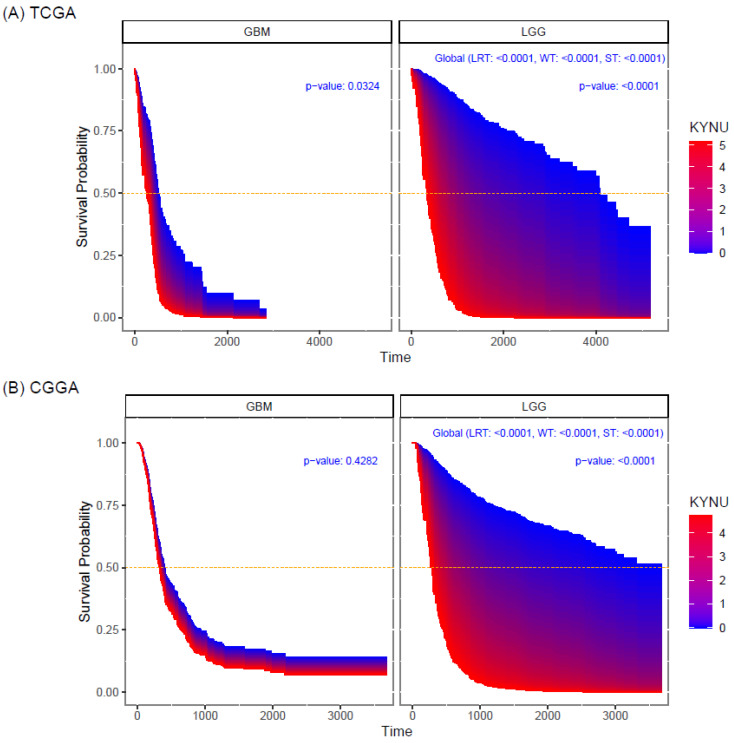
Survival area plots for patients diagnosed with GBM and LGG according to the expression levels of KYNU (log-2 scale). Survival probability of (**A**) TCGA samples and (**B**) CGGA samples. The survival probability curves are obtained from a Cox proportional hazards regression model.

**Figure 6 pharmaceuticals-16-00369-f006:**
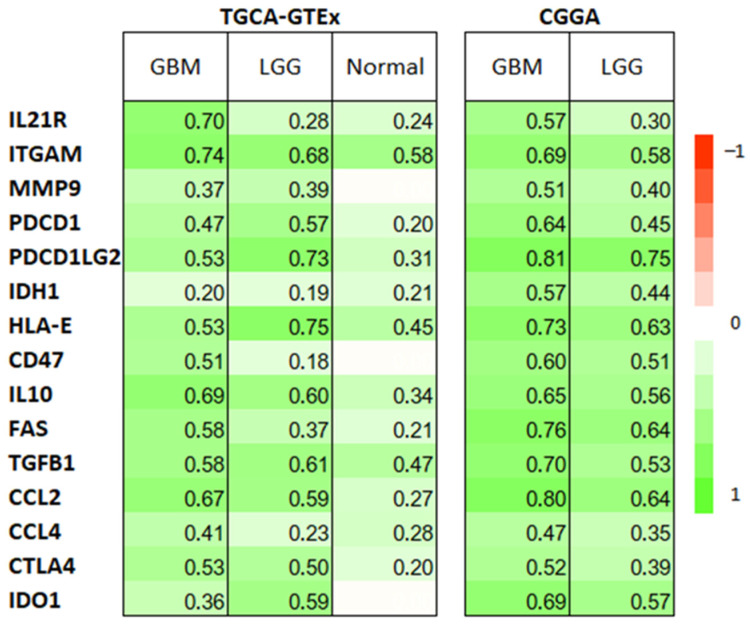
Heat map of Spearman’s correlations between KYNU expression and the expression of genes associated with immune response. Only the significant pairwise correlations were considered, using the Holm adjustment for multiple tests in each group of samples (GBM and LGG and Normal).

**Figure 7 pharmaceuticals-16-00369-f007:**
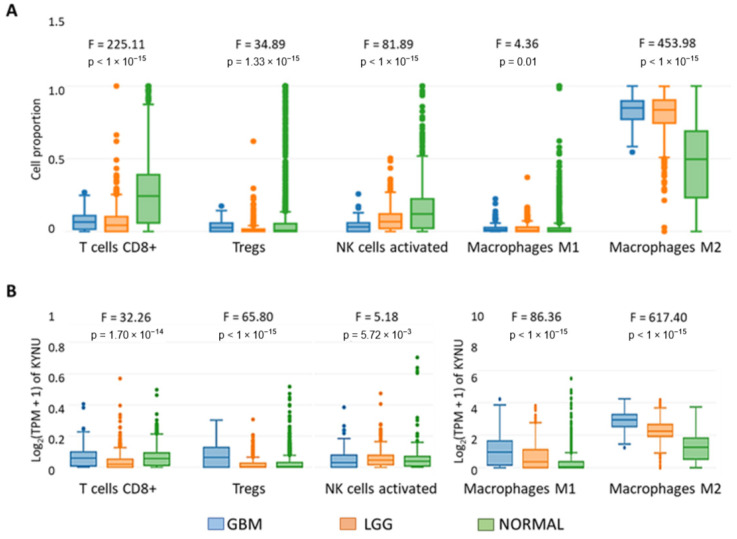
Infiltration level of immune cells: T cells CD8+, Tregs, activated NK cells, macrophages M1 and macrophages M2. For each cell type, the boxplots represent (**A**) its proportional distribution in each bulk sample (with normalization) and (**B**) the KYNU expression distribution, considering the samples associated with GBM (TCGA), LGG (TCGA) and Normal (GTEx-Brain). The *p*-values refer to the one-way ANOVA to compare GBM, LGG and Normal.

**Figure 8 pharmaceuticals-16-00369-f008:**
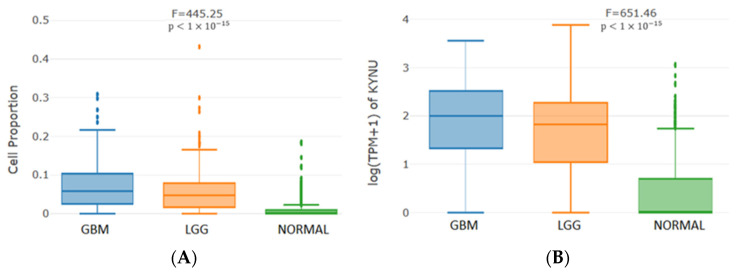
Infiltration level of myeloid dendritic cells: (**A**) its proportional distribution in each bulk sample (without normalization) and (**B**) the KYNU expression distribution, considering the samples associated with GBM (TCGA), LGG (TCGA) and Normal (GTEx-Brain). The *p*-values correspond to the one-way ANOVA to compare GBM, LGG and Normal.

## Data Availability

The datasets generated and/or analyzed during the current study are available in public databases.

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
