# Peer review of "Kynureninase Promotes Immunosuppression and Predicts Survival in Glioma Patients: In Silico Data Analyses of the Chinese Glioma Genome Atlas (CGGA) and of the Cancer Genome Atlas (TCGA)"

_pharmaceuticals, 2023, doi:10.3390/ph16030369_

Round 1

Reviewer 1 Report

In this study, the authors used in silico data analysis approach and investigated the clinical relevance of Kynureninase in glioma patients by analyzing gene expression data in the Chinese Glioma Genome Atlas (CGGA) and of The Cancer Genome Atlas (TCGA). This study is well-designed and comprehensive in terms of clinical relevance. My few concerns are as follows:

1.       Correlation between KYNU and indoleamine 2,3-dioxygenase 1 (IDO1) should be analyzed because it is an important immunosuppressive molecule in the Kynurenine pathway.

2.       Infiltration level of MDSC should be included in Fig.6.

Reviewer 2 Report

In their paper “Kynureninase promotes immunosuppression and predicts survival in glioma patients: in silico data analyses of Chinese Glioma Genome Atlas (CGGA) and of The Cancer Genome Atlas (TCGA)”, Pérez de la Cruz et al. strive to explore the role of kynureninase in glioma and as a potential therapeutic target. Through computational analyses, the authors observe a correlation between KYNU expression and overall survival as well as immune response, a finding that has the potential to impact targeted therapies for brain cancers. Apart from the comments below, this paper has technically sound findings and the data support the authors’ conclusions.

Major Points

1)      Would be helpful to add a figure or more thorough explanation of the kynurenine pathway. It will aid in readers’ understanding of the role of KYNU, KYN, IDO1, KYNA, 3-HK, 3-HANA, and KMO, as well as explain what the short and long arms of this pathway means.

2)      The body map in Figure 1A is hard to read without a color scale. It is hard to determine differences between tumor and normal tissue KYNU expression unless looking only at the table, in which case the body map is unnecessary.

3)      In line 155, IDH mutation status is mentioned for the first time. This could have a major impact on different metabolic states for different tumors – the relevance of this should be explained, and maybe earlier in the paper.

4)      In section 2.4, can the immune mediators’ expression be compared to normal tissue?

5)      While this is a thorough computational analysis, it could be stronger with some experimental validation to follow up on the immune findings.

Minor Points

1)      Some grammatical errors highlighted in green

2)      While the introduction and discussion focus on GBM, many of the significant findings are more specific to LGG. Including LGG in these sections is also important.

Reviewer 3 Report

I hope the authors consider these comments:

1.       The authors should design a schematic figure in the introduction section representing the KP and related pathways and metabolites. Besides, a concept map of these pathways is of interest.

2.       In figure 1A, you should sort the cancers in the table and chart based on the KYNU expression in tumor tissues.

3.       In the body map of figure1, which body relates to the tumor or healthy tissues? It seems your color selection is hard to follow.

4.       There are some abbreviations without extended forms, such as IDH.

5.       The manuscript needs complete revision considering many grammatical mistakes and syntax errors.

Round 2

Reviewer 2 Report

The authors have addressed all comments quite sufficiently. There is a typo in line 305 (new in this version). This paper is ready for publication.